

**Aerosol trace element solubility determined using ultrapure water batch leaching: an**
**intercomparison study of four different leaching protocols**
Rui Li,[1,2] Prema Piyusha Panda,[3,4] Yizhu Chen,[2,5] Zhenming Zhu,[6] Fu Wang,[6] Yujiao Zhu,[7]
He Meng,[8] Yan Ren,[6] Ashwini Kumar,[3,4] Mingjin Tang[2,5,*]
[1] Department of Environmental Health, School of Public Health, Shanxi Medical University,
Taiyuan, China
[2] State Key Laboratory of Organic Geochemistry, Guangzhou Institute of Geochemistry,
Chinese Academy of Sciences, Guangzhou, China
[3] CSIR-National Institute of Oceanography, Dona Paula, Goa, India
[4] School of Earth, Ocean and Atmospheric Sciences, Goa University, Goa, India
[5] College of Earth and Planetary Sciences, University of Chinese Academy of Sciences,
Beijing, China
[6] Longhua Center for Disease Control and Prevention of Shenzhen, Shenzhen, China
[7] Environment Research Institute, Shandong University, Qingdao, China
[8] Qingdao Eco-environment Monitoring Center of Shandong Province, Qingdao, China
*Correspondence: Mingjin Tang (mingjintang@gig.ac.cn)



**Abstract**

Solubility of aerosol trace elements, which determines their bioavailability and reactivity, is
operationally defined and strongly depends on the leaching protocol used. Ultrapure water
batch leaching is one of the most widely used leaching protocols, while the specific leaching
protocols used in different labs can still differ in agitation methods, contact time, and filter pore
size. It is yet unclear to which extent the difference in these experimental parameters would
affect the aerosol trace element solubility reported. This work examined the effects of agitation
methods, filter pore size and contact time on the solubility of nine aerosol trace elements, and
found that the difference in agitation methods (shaking vs. sonication), filter pore size (0.22 vs.
0.45 μm), and contact time (1 vs. 2 h) only led to small and sometimes insignificant difference
in the reported solubility. We further compared aerosol trace element solubility determined
using four ultrapure water leaching protocols which are adopted by four different labs and vary
in agitation methods, filter pore size and/or contact time, and observed good agreement in the
reported solubility. Therefore, our work suggests that although ultrapure water batch leaching
protocols used by different labs vary in specific experimental parameters, the determined
aerosol trace element solubility is comparable. We recommend ultrapure water batch leaching
to be one of the reference leaching schemes, and emphasize that additional consensus in the
community on agitation methods, contact time and filter pore size is needed to formulate a
standard operating procedure for ultrapure water batch leaching.




# 1 Introduction

Aerosol trace elements, originating from natural and anthropogenic sources, are of great

concerns, as they significantly impact marine and terrestrial ecosystems (Boyd and Ellwood,

2010; Dong et al., 2023; Mahowald et al., 2018), have adverse effects on human health

(Dahmardeh Behrooz et al., 2021; Fang et al., 2017; Gao et al., 2022; Wei et al., 2019), and

play important roles in atmospheric chemistry (Al-Abadleh, 2024; Alexander et al., 2009; Mao

et al., 2013; Martin and Hill, 1987; Wang et al., 2021). The dissolved fraction of aerosol trace

elements, instead of their total abundance, is considered to be bioavailable (Baker and Croot,

2010; Ito et al., 2012; Mukhtar and Limbeck, 2013) and more chemically reactive in the

atmosphere (Kebede et al., 2016; Mao et al., 2017). Dissolved trace elements are typically

referred to as the fraction of elements which can pass through a filter with certain pore size

(usually 0.2-0.22 or 0.45 μm) after aerosol particles are dissolved in certain aqueous solutions

(Boyd and Ellwood, 2010; Ito and Xu, 2014; Meskhidze et al., 2016; Myriokefalitakis et al.,

2018). Solubility (or fractional solubility, to be more precise), which is defined as the ratio

(in %) of the dissolved element to the total element (Baker et al., 2006; Sholkovitz et al., 2012),

largely determines the bioavailability and reactivity of aerosol trace elements.

A wide range in the solubility has been reported in the literature for a given trace element

in atmospheric aerosols, and for example, the reported solubility of aerosol Fe ranges from <1%

to >90% (Baker and Jickells, 2006; Sholkovitz et al., 2012). Such wide variabilities in aerosol

trace element solubility, on one hand, can be caused by difference in sources and aging

processes of aerosol particles examined (Ito et al., 2021; Meskhidze et al., 2019); on the other

hand, they also stem from various leaching protocols which were used by different studies



(Chen et al., 2006; Li et al., 2023; Upadhyay et al., 2011).
Various leaching protocols have been used in previous studies to extract dissolved aerosol
trace elements, as summarized in a recent paper (Li et al., 2023). In brief, available leaching
protocols broadly consist of two catalogues, including flow-through leaching and batch
leaching. Flow-through leaching is instantaneous and typically has a contact time (between
aerosol particles and the leaching solution used) of tens of seconds, and batch leaching usually
has a much longer contact time (tens of minutes or longer). Compared to flow-through leaching,
batch leaching is more widely used in atmospheric research. Furthermore, for batch leaching,
various leaching solutions were used in previous studies, such as ultrapure water, filtered
seawater, and formate/acetate buffers. Compared to filtered seawater and formate/acetate
buffers, ultrapure water is more widely used in atmospheric research due to its simplicity and
reproducibility (Li et al., 2023; Meskhidze et al., 2019); another important reason is that
ultrapure water leaching does not introduce any other chemical species (except water) and thus
can simultaneously extract water-soluble ions and organics for additional analysis.
Even for ultrapure water batch leaching, protocols used by different studies may still vary
in agitation methods, contact time, and filter pore size; nevertheless, the effects of these factors
on the reported solubility are not well understood. First, some labs use sonication to agitate the
leaching solutions (Chen et al., 2006; Kumar and Sarin, 2010; Liu et al., 2021; Longo et al.,
2016; Shi et al., 2020), and other labs use shaking (Baker et al., 2003; Gao et al., 2020; Hsu et
al., 2010; Li et al., 2022; Salazar et al., 2020). Sonication may cause changes in chemical
composition and formation of reactive oxygen species in the solution (Juretic et al., 2015;
Miljevic et al., 2014); however, it remains to be examined whether sonication will change the





solubility of aerosol trace elements. Second, filters with different pore sizes, including 0.2-0.22
and 0.45 μm (and 0.02 μm to a less extent), are employed to filter the leaching solutions,
contributing to the uncertainties in the reported solubility; however, the effects of filter pore
size have seldom been experimentally examined. Third, some studies (Li et al., 2023; Mackey
et al., 2015) suggested that contact time (2-8 h) could also influence the reported solubility.

In the present work, using aerosol particles collected at a suburban site close to the

coastline of the Northwest Pacific, we investigated to which extent different ultrapure water
batch leaching protocols would affect reported aerosol trace element solubility. In the first part
of this work, we examined the effects of agitation (shaking vs. sonication), filter pore size (0.22
vs. 0.45 μm) and contact time (1 vs. 2 h) on the reported solubility of nine aerosol trace elements.
In the second part, we compared solubility determined using protocols commonly adopted by
four labs. The four labs all use ultrapure water batch leaching, but the leaching protocols they
use differ in agitation method, contact time, and/or filter pore size.
**2 Experimental section**
**2.1 Sample collection and distribution**
We collected aerosol samples between 18 March and 22 April 2023 in Qingdao, a coastal
city in northern China, typically impacted by desert dust and anthropogenic aerosols in spring.
As described elsewhere (Zhang et al., 2022), aerosol sampling took place at a suburban site
which was about 1.3 km from the coast. A cumstom-made high-volume aerosol sampler (ASM-
1; flow rate: 1 $m^3$/min) was deployed on a building roof (about 20 m above the ground) to
collect $PM_{10}$ samples. Aerosol sampling started at 08:00 am each day and stopped at 07:30 am
on the next day, resulting in a sampling volume of 1410 $m^3$. $PM_{10}$ samples were collected onto





pre-cleaned Whatman 41 cellulose fiber filters (25 cm × 20 cm) which had very low
backgrounds for trace elements (Morton et al., 2013; Zhang et al., 2022). In total we collected
26 filter samples, 4 sampling blanks, and 3 lab blanks: lab blanks were defined as pre-cleaned
filters, and sampling blanks were defined as pre-cleaned filters which were placed in the aerosol
sampler for 2 h when the sampling flow was off.
After aerosol collection, each filter was folded inward and placed into a clean
polyethylene bag (12 inch × 9 inch, supplied by Sigma-Aldrich) which was used due to its low
background (Morton et al., 2013), and then stored at -20 °C. A titanium punch was used to
obtain 10 subsamples (47 mm in diameter) from each filter sample, and these subsamples were
stored at -20 °C.
**2.2 Measurement of total and dissolved trace elements**
**2.2.1 Total elements**
As shown in Table 1 and described below, for the 10 subsamples obtained from each
original filter sample, the first subsample was digested to determine total elements, another
eight subsamples were leached using different protocols to determine dissolved elements, and
the last subsample was reserved for any unforeseen purpose (but was not used at the end).
Subsample 1 was digested in a Teflon jar which contained a mixture of $HNO_3$-$HF$-$H_2O_2$,
using a microwave digestion system (Zhang et al., 2022). After digestion, we evaporated
residual acids in the Teflon jar, and filled it with 20 mL $HNO_3$ (1%). Subsequently, we filtered
the solution through a polyethersulfone membrane syringe filter (with a pore size of 0.22 μm),
and then used ICP-MS (inductively coupled plasma mass spectrometry, iCAP Q, Thermo
Fisher) to measure nine trace elements, including Fe, Al, As, Cr, Cu, Mn, Pb, V and Zn.




**Table 1.** Overview of protocols used to digest and leach subsamples examined in this work.

| subsample | agitation | contact time (h) | filter pore size (μm) | Lab | References |
|---|---|---|---|---|---|
| 1 | digestion | | | GIG | |
| 2a | shaking | 2 | 0.22 | GIG | Zhang et al. (2022) |
| 2b | shaking | 2 | 0.22 | GIG | |
| 2c | shaking | 2 | **0.45** | | |
| 2d | **sonication** | 2 | 0.22 | | |
| 2e | shaking | **1** | 0.22 | | |
| 3a | sonication | 1 | 0.22 | ZJU | Liu et al. (2021) |
| 3b | sonication | 1 | 0.45 | OUC | Shi et al. (2020) |
| 3c | sonication | 0.5 | 0.20 | NIO | Panda et al. (2022) |


### 2.2.2 Dissolved elements

Subsample 2a was leached using the protocol adopted by the lab at Guangzhou Institute

of Geochemistry (GIG) (Li et al., 2023; Zhang et al., 2022; Zhang et al., 2023). In brief, the

subsample was shredded and then immersed in 20 mL ultrapure water for 2 h, stirred using an

orbital shaker; subsequently, the solution was filtered through a polyethersulfone membrane

syringe filter (with a pore size of 0.22 μm). After that, the solution was immediately acidified

with a small volume of high-purity $HNO_3$ to contain 1% $HNO_3$, and then analyzed using ICP-

MS to determine dissolved trace elements. Subsample 2b was leached using the same protocol

as subsample 2a, and the purpose was to examine whether aerosol particles were

homogeneously distributed on different subsamples, and to assess the repeatability of the GIG

leaching protocol.



Subsamples 2c-2e were leached using protocols similar to that used to leach subsample
2a. As summarized in Table 1, the only difference to the protocol used to leach subsample 2a
was the filter pore for 2c (0.45 μm, vs. 0.22 μm for 2a), agitation method for 2d (sonication, vs.
shaking for 2a), and contact time for 2e (1 h, vs. 2 h for 2a). The purpose of using subsamples
2c-2e is to examine the effects of filter pore size (0.22 vs. 0.45 μm), agitation method (shaking
vs. agitation), and contact time (1 vs. 2 h) on the reported solubility.
Subsamples 3a, 3b and 3c were leached using the protocols typically used by ZJU
(Zhejiang University, China), OUC (Ocean University of China, China) and NIO (National
Institute of Oceanography, India), respectively, in order to compare solubility determined by
the GIG lab with those reported by the other three labs. Please note that subsamples 3c were
leached and analyzed by NIO, while subsamples 3b and 3c were leached and analyzed at GIG
(using the ZJU and OUC protocols, respectively).
Subsample 3a was leached at GIG using the ZJU protocol (Liu et al., 2021; Zhu et al.,
2020). In brief, each subsample was shredded and immersed in 20 mL ultrapure water, and the
aqueous mixture was sonicated for 1 h during which the water bath temperature was kept below
30 °C; after that, the aqueous mixture was filtered using a polyethersulfone membrane syringe
filter (pore size: 0.22 μm) and acidified for later ICP-MS analysis. Subsample 3b was leached
at GIG using the OUC protocol (Shi et al., 2020), which is very similar to the ZJU method: the
only difference is that the filter pore size was 0.45 μm for the OUC protocol and 0.22 μm for
the ZJU protocol.
Subsample 3c was leached and analyzed at NIO using the NIO protocol (Panda et al.,
2022). In brief, each subsample was shredded and placed into a pre-cleaned Savillex vial (50





mL); after that, the vial was filled with 20 mL ultrapure water, caped, and then sonicated for
30 minutes to agitate the aqueous mixture (but in 2 cycles with 15 min for each cycle, in order
to maintain the water bath at room temperature). The aqueous mixture was then filtered through
a Whatman PVDF syringe filter (pore size: 0.2 μm), and then acidified with $HNO_3$ (2% v/v)
for later high-resolution ICP-MS analysis (Nu Instruments, Attom ES).
**3 Results and discussion**

Subsamples 2a and 2b were identically leached using the protocol GIG normally uses, and

the paired $t$-test ($\alpha = 0.05$) was employed to examine whether the difference in obtained
solubility was significant. As summarized in Table 2, the difference in obtained solubility was
not statistically significant between 2a and 2b for Fe, Al, As, Mn, Pb, and V; furthermore,
Figure S1 suggests good linear correlations in solubility between 2a and 2b for the six elements
(R > 0.99), and the corresponding slopes (0.98 to 1.02) were very close to 1. For the other three
elements (Cr, Cu and Zn), although the difference in solubility was found to be statistically
significant between 2a and 2b (Table 1), good linear correlations between the solubility were
found (R > 0.97) and the slopes (1.00-1.06) were close to 1; therefore, the difference in
solubility between 2a and 2b, if it existed, was small for Cr, Cu and Zn.

In summary, we conclude that the distribution of aerosol particles on a given original filter

was homogeneous and that the protocol GIG normally uses had very good repeatability.

**Table 2.** Summary of statistical analysis (paired-$t$-test, $\alpha = 0.05$) which examined whether the
difference in solubility obtained for different groups of subsamples is statistically significant.
Solubility obtained for subsamples 2a is compared with those obtained for subsamples 2b, 2c,



2d and 2e, respectively. Y: the difference is statistically different; N: the difference is not
statistically different.

| element | 2a vs. 2b | 2a vs. 2c | 2a vs. 2d | 2a vs. 2e |
|---|---|---|---|---|
| Fe | N | **Y** | **Y** | **Y** |
| Al | N | N | N | **Y** |
| As | N | N | **Y** | **Y** |
| Cr | **Y** | **Y** | **Y** | **Y** |
| Cu | **Y** | **Y** | **Y** | **Y** |
| Mn | N | N | **Y** | **Y** |
| Pb | N | **Y** | **Y** | N |
| V | N | N | N | N |
| Zn | **Y** | **Y** | **Y** | **Y** |


### 3.1 The effects of filter pore size

To examine the effects of filter pore size on the reported solubility, subsamples 2a and 2c
were leached using very similar protocols, and the only difference is the pore size (2a: 0.22 μm;
2c: 0.45 μm) of filters used (Table 1).
The difference in obtained solubility was not statistically significant between 2a and 2c
for Al, As, Mn and V (Table 2); moreover, good linear correlations between 2a and 2c were
found for the four elements (Figure 1), with slopes (0.96-1.04) very close to 1. For the other
five elements (Fe, Cr, Cu Pb, and Zn), the difference in solubility between 2a and 2c was found
to be statistically significant; however, solubility between 2a and 2c was very well linearly
correlated (Figure 1), with slopes (1.03-1.12) close to or slightly larger than 1.
To conclude, among the nine elements we examined, the effects of filter pore size (0.22
vs. 0.45 μm) on reported solubility were found to be insignificant for four elements (Al, As,
Mn and V) and very smaller for the other five elements (Fe, Cr, Cu, Pb and Zn).

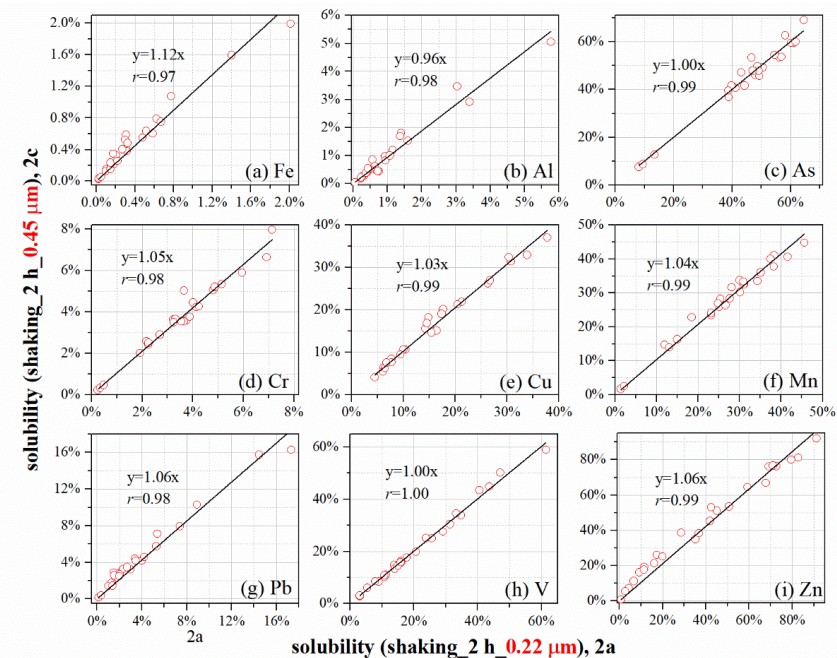


**Figure 1.** The effects of filter pore size (2a: 0.22 μm; 2c: 0.45 μm) on measured element

solubility. The only difference in protocols used to leach subsamples 2a and 2c is the filter pore

size (0.22 versus 0.45 μm).


**3.2 The effects of agitation**

     As shown in Table 1, the only difference between the protocol used to leach subsamples

2a and that used to leach subsamples 2d is the agitation method used (2a: shaking; 2d:

sonication), and solubility obtained for subsamples 2a and 2d was compared to assess the

effects of agitation methods on reported solubility.

     Table 2 shows that the reported solubility between 2a and 2d was not statistically different

for two elements (Al and V); in addition, good linear correlations between 2a and 2d were



found for the two elements (Figure 2), and these slopes (1.10 for Al and 1.05 for V) were quite
close to 1. With respect to the other seven elements (Fe, As, Cr, Cu, Mn, Pb and Zn), on one
hand, the difference in solubility between 2a and 2d was found to be statistically significant;
on the other hand, good linear correlations in solubility existed between 2a and 2d (Figure 2),
and these slopes were in the range of 0.94-1.21.

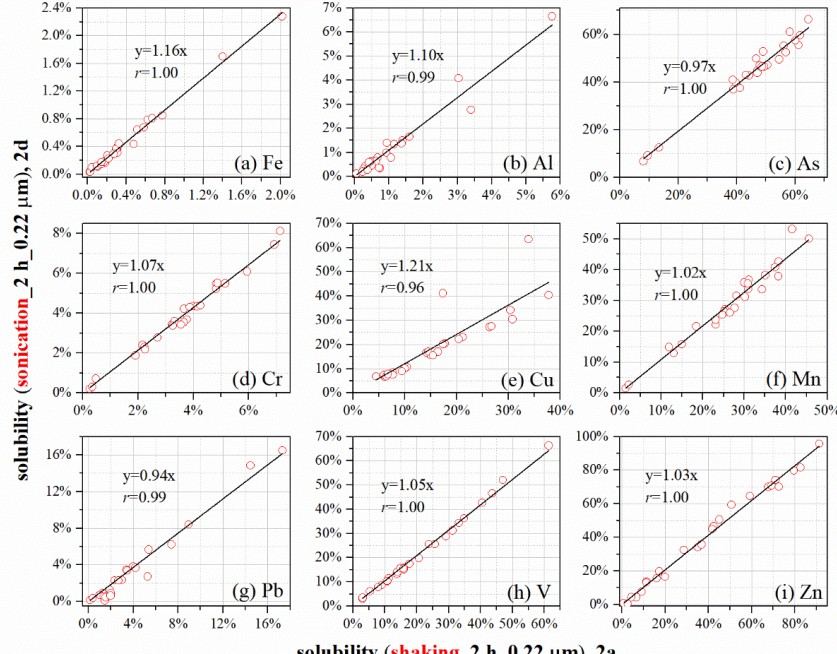


**Figure 2.** The effects of agitation (2a: shaking; 2d: sonication) on measured element solubility.
The only difference in protocols used to leach subsamples 2a and 2d is the agitation method
(shaking vs. sonication).

In summary, we found that the choice of agitation methods (shaking vs. sonication) had
no measurable (for Al and V) or small effects (for Fe, As, Cr, Cu, Mn, Pb and Zn) on the
reported solubility.





### 3.3 The effects of contact time


To assess the impacts of contact time on reported solubility, subsamples 2a and 2e were
leached using very similar protocols, and the only difference was contact time (2a: 2 h; 2e: 1 h).
As shown in Table 2, the reported solubility were not statistically different between 2a
and 2e for Pb and V; moreover, good linear relationships between 2a and 2e were found for the
two elements (Figure 3), with slopes close to 1 (1.24 for Pb and 1.02 for V). For the other seven
elements, their solubility was found to be statistically significant between 2a and 2e (Table 2);
nevertheless, for each of the seven elements, the solubility reported for 2a was very well
linearly correlated with that reported for 2e (Figure 3), and the slopes were close to 1 (in the
range of 0.95-1.28).

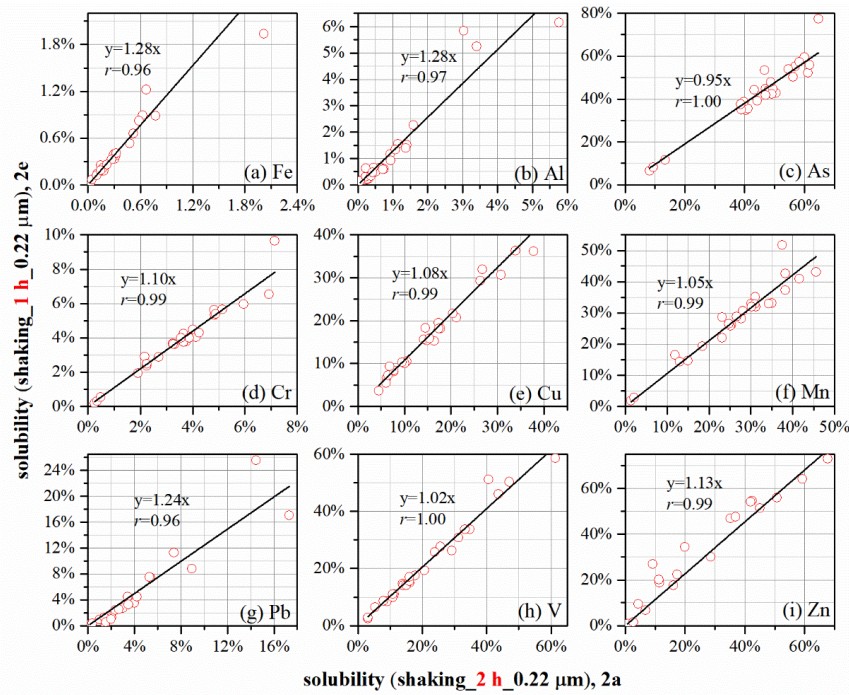






**Figure 3.** The effects of contact time (2a: 1 h; 2e: 2 h) on measured element solubility. The only difference in protocols used to leach subsamples 2a and 2e is the contact time (1 h versus 2 h).

To summarize, our present work suggested that the increase in contact time from 1 h to 2 h would cause insignificant or small effects on reported solubility. Using a different set of aerosol samples, our previous work (Li et al., 2023) compared the measured solubility obtained with longer contact time (4 and 8 h) to that obtained with a contact time of 2 h. As shown in Table S1, increase in contact time from 2 to 4 h would cause significant increase in solubility, on average by a factor of ~1.3 for Zn to ~3.1 for As (Li et al., 2023). It is still not clear why the increase in contact time from 1 to 2 h would not cause significant change in aerosol trace element solubility while the increase in contact time from 2 to 4 h would.

**3.4 Comparison of solubility obtained using protocols commonly used by four labs**

We further compared solubility determined using the GIG protocol with those determined using ZJU, OUC and NIO protocols, respectively. Table 3 summarizes the slopes obtained from correlation analysis (Figures 4 and S2-S8). The NIO lab measured eleven elements, among which five elements (Fe, Al, Cu, Mn, and V) were measured using the other three protocols; as a result, the solubility of these five elements determined using the NIO protocol was compared with those determined using the GIG protocol.

**Table 3.** Correlations between solubility determined using the GIG protocol and that determined using ZJU, OUC and NIO protocols. Here only the slopes ($k$) are provided.



|  | ZJU | OUC | NIO |
|---|---|---|---|
|  | k | k | k |
| Fe | 1.03 | 1.39 | 1.82 |
| Al | 1.09 | 1.19 | 1.80 |
| As | 0.96 | 0.87 | -- |
| Cr | 0.96 | 0.97 | -- |
| Cu | 1.06 | 1.04 | 0.99 |
| Mn | 0.99 | 0.98 | 1.09 |
| Pb | 0.97 | 1.01 | -- |
| V | 1.06 | 0.99 | 1.05 |
| Zn | 0.97 | 0.94 | -- |


With respect to Fe solubility, GIG results were very well correlated with ZJU results ($r$ =
0.99, Figure 4a) and the slope was found to be 1.03, suggesting good agreement between GIG
and ZJU; GIG results were also well correlated with while overall larger than OUC results ($r$
= 0.98, Figure 4b), and the slope was determined to be 1.39; good correlation was also found
between GIG and NIO results ($r$ = 0.91, Figure 4c), and the slope was determined to be 1.82,
indicating that Fe solubility determined using the NIO protocol was larger than that determined
using the GIG protocol. Similarly, with respect to Al solubility (Figure S2 and Table 3), the
GIG results were well correlated with ZJU, OUC and NIO results, and correlations were best
for ZJU ($r$ = 0.99) and moderate for NIO ($r$ = 0.93); in addition, the slopes were determined to
be 1.09, 1.19 and 1.80 for ZJU, OUC and NIO results, respectively.
With respect to Cu (Figure S5), Mn (Figures 4d-4f) and V (Figure S7), their solubility
determind using the ZJU, OUC and NIO protocols was well correlated with that determined
using the GIG protocol, and the slopes obtained from correlation analysis, which ranged from
0.98 to 1.09 (Table 3), were all close to 1.
Since As, Cr, Pb and Zn were not measured using the NIO protocol, we only compared
GIG results with ZJU and OUC results for these four elements. As shown in Figures S3, S4,



S6 and S8, the solubility of As, Cr, Pb and Zn determined using ZJU and OUC protocols was
well correlated with that determined using the GIG protocol, and the slopes (0.87-1.01, as
summarized in Table 3) were close to 1.

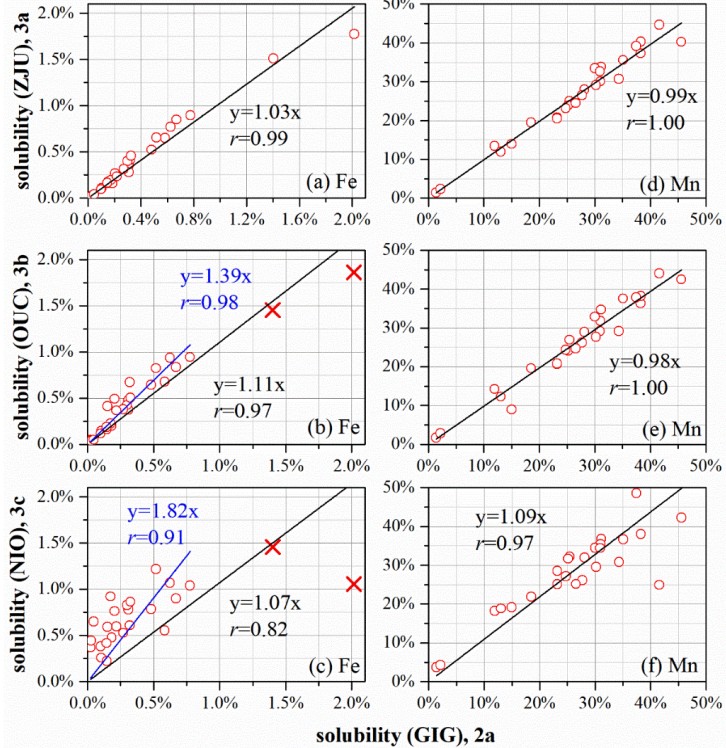


**Figure 4.** Solubility of Fe and Mn determined using the GIG protocol vs. those determined

using ZJU, OUC and NIO protocols: (a) Fe, GIG vs. ZJU; (b) Fe, GIG vs OUC; (c) Fe, GIG

vs. NIO; (d) Mn, GIG vs. ZJU; (e) Mn, GIG vs OUC; (f) Mn, GIG vs. NIO.


To summarize, although the four ultrapure water batch leaching protocols differ in
agitation method, contact time and/or filter pore size (GIG: shaking, 2 h contact time, 0.22 μm
filter pore size; ZJU: sonication, 1 h contact time, 0.22 μm filter pore size; OUC: sonication, 1
h contact time, 0.45 μm filter pore size; NIO: sonication, 1 h contact time, 0.22 μm filter pore





size), for the nine elements examined in this intercomparison study, their solubility determined
using the four protocols in general showed good agreement. This is consistant with the results
presented in Sections 3.1-3.3, where we found that the effects of agitation method (shaking vs.
sonication), contact time (1 vs. 2 h) and filter pore size (0.22 vs. 0.45 μm) were rather limited.
The solubility of Fe and Al determined using the NIO protocol deviated considerably from
those determined using the GIG protocol, probably because Fe and Al solubility was very low
(mostly <2%) and small change in leaching protocol may cause significant change in the
amounts of Fe and Al dissolved.

## 301   4 Conclusion

Ultrapure water batch leaching is widely used in atmospheric research to determine
aerosol trace element solubility, and the specific leaching protocols used in different labs can
still vary in agitation methods, contact time, and filter pore size. It is yet unclear to which extent
the difference in these experimental parameters would affect the reported aerosol trace element
solubility; in other words, it remains to be examined whether solubility reported by previous
studies which used different ultrapure water batch leaching protocols is comparable.
We examined the effects of agitation methods, filter pore size and contact time on the
reported solubility of nine aerosol trace elements, including Fe, Al, As, Cr, Cu, Mn, Pb, V and
Zn. It was found that the difference in agitation methods (shaking vs. sonication), filter pore
size (0.22 vs. 0.45 μm), and contact time (1 vs. 2 h) only led to small and sometimes
insignificant difference in the reported solubility. We further compared aerosol trace element
solubility determined using four widely used ultrapure water leaching protocols which differ
in agitation methods, filter pore size and/or contact time, and in general the solubility



determined using the four protocols was found to be in good agreement. Therefore, aerosol
trace element solubility determined in previous studies using ultrapure water batch leaching
may be comparable.

Aerosol trace element solubility is an operationally defined term (Baker and Croot, 2010;

Meskhidze et al., 2019), and strongly depends on the leaching protocol employed. A number
of leaching protocols have been used in previous studies to extract dissolved trace elements,
making it very challenging to compare solubility reported in different studies (Perron et al.,
2024). In order to reduce uncertainties in aerosol trace element solubility, it is necessary to
formulate standard operating procedures for frequently-used aerosol leaching protocols. Our
current work suggests that although ultrapure water batch leaching protocols used by different
labs vary in specific experimental parameters, the determined aerosol trace element solubility
showed good agreement; furthermore, ultrapure water batch leaching is operationally simple
and does not introduce any other chemical species which may interfer analysis of water-soluble
inorganic ions and organics. Therefore, we recommend ultrapure water batch leaching to be
one of the reference leaching schemes. Before a standard operating procedure can be
formulated for ultrapure water batch leaching, the community will need to reach consensus on
agitation methods, contact time and filter pore size, and further intercomparison studies,
preferentially with more labs involved, will be very helpful.

**Data availability.**
Data are available upon request (Mingjin Tang: mingjintang@gig.ac.cn).
**Author contributions.**





Rui Li: methodology, formal analysis, investigation, writing - original draft, writing - review
& editing; Prema Piyusha Panda: formal analysis, investigation; Yizhu Chen: investigation;
Zhenming Zhu: investigation; Fu Wang: investigation, supervision; Yujiao Zhu: resources; He
Meng: resources; Yan Ren: resources, supervision; Ashiwini Kumar: resources, writing -
review & editing, supervision; Mingjin Tang: conceptualization, methodology, resources,
writing - original draft, writing - review & editing, supervision.
**Competing interests.**
Mingjin Tang is a member of the editorial board of Atmospheric Measurement Techniques.
**Acknowledgements.**
The authors would like to thank Professor Likun Xue and other colleagues at Shandong
University for their assistance in aerosol sampling.
**Financial support.**
This work was supported by the National Natural Science Foundation of China (42321003 and
42277088), Guangzhou Bureau of Science and Technology (2024A04J6533), and Guangdong
Basic and Applied Basic Research Foundation (2022A1515110371).






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
