# Peer review of "Aerosol trace element solubility determined using ultrapure water batch leaching: an"

_Atmospheric Measurement Techniques, 2024_

## Referee Comment (RC3)

Li et al. used several ultrapure water batch leaching protocols to examine the effects of agitation methods, contact time, and filter pore size on the solubility of trace elements. This is good work and is recommended to be published in AMT after concerning the following weaknesses.

**Major comments:**

1. Why is it needed to "formulate a standard operating procedure for ultrapure water batch leaching" when agitation methods, contact time, and filter pore size led to small or even insignificant differences in the solubility of trace elements?

2. In lines 246-253, the authors mentioned that longer contact time (2 and 4h) would cause an increase in solubility by 1.3 times for Zn and ~3.1 for As. However, why does this study only consider the contact time of 0.5-2h?

3. More comparative experiments should be added to avoid uncertainties in each group for testing the distribution of aerosol particles and the effects of agitation methods, contact time, and filter pore size.

**Minor suggestion**

1. Add the representation of the circles within each figure or in the first figure.

---

## Author Comment (AC1)

Comments by referees are in blue.

Our replies are in black.

Changes to the manuscript are highlighted in red both here and in the revised manuscript.

**Reply to referee #1**

The manuscript by Li et al. summarized batch leaching protocols affecting aerosol trace element solubility. They compared three items, namely, agitation methods, contact time, and filter pore size to identify the difference in these experimental parameters would affect the aerosol trace element solubility. Moreover, they also compared solubility determined using protocols commonly adopted by four labs (GIG, ZJU, OUC and NIO). Their laboratory research helps to understand the impact of different processing methods on the solubility of metal elements. After addressing the following minor issues, I recommend their research to be published.

**Reply:** We would like to recommend ref #1 for reviewing our manuscript and recommending it for publication. We have addressed his/her comments which greatly help us improve our manuscript, and revised the manuscript accordingly, as detailed below.

Number of samples should be added in each subgroup in Table 1.

**Reply:** Although such information is provided in our original manuscript (page 6 line 111-114), as suggested by the referee, we have included a sentence in the caption of Table 1 in the revised manuscript (page 7) to provide relevant information: "For each protocol, 26 subsamples were examined."

There are two points in Figure 2e that show a clear large difference in the solubility of Cu obtained using the two methods, why?

**Reply:** Indeed there are two points which deviate significantly from the overall trend. This may be caused by contamination during experimental processes. As the two points does not affect the overall result, we choose not to discuss them in specific.

As can be seen from the Figure 3, why element solubility conducted by the contact time from 1 to 2 h higher than that from 2 to 4 h?

**Reply:** Increase in solubility was larger when contact time was increased from 2 h to 4 h than that when contact time was increased from 1 h to 2 h. As we stated in the original manuscript (page 14, line 249-253), the underlying reason is unclear, and one possible reason is that for a given element, different speciation have different dissolution kinetics. In response to this comment, we

have added one sentence to provide possible explanation in the revised manuscript (page 14-15): "It is still not clear why the increase in contact time from 1 to 2 h would not cause significant change in aerosol trace element solubility while the increase in contact time from 2 to 4 h would, probably because for a given element, different speciation have different dissolution kinetics."

What do the blue lines and texts in Figure 4 represent? What does the red cross represent?

**Reply:** In response to this comment, we have added the following sentence in the figure caption in the revised manuscript (page 17) to provide necessary explanation: "Black lines represent fitting when all the data points are included, and blue lines represent fitting when outliers (represented by red crosses) are excluded." In addition, similar changes are implemented for figures in SI when necessary.

In this study, the authors only compared the differences between the different pre-treatments, not the differences measured by the different instruments, so there were no significant differences between methods (mainly pre-treatments), which might not be the case if they use different instruments (ICP-MS VS. ferrozine technique methods). Please add a short discussion to discuss this issue and refer to some of the literature, e.g. Zhu et al. 2022, 22 2191–2202.

**Reply:** We agree with the referee. Accordingly, in the revised manuscript (page 19) we have included the following sentence to discuss this aspect: "Trace elements were analyzed using similar methods (ICP-MS) in our present work and thus essentially we only examined the effects of leaching protocols; nevertheless, other methods were also used by some previous studies to measure trace elements (Fang et al., 2015; Zhu et al., 2022), probably causing additional uncertainties."

---

## Author Comment (AC2)

Comments by referees are in blue.

Our replies are in black.

Changes to the manuscript are highlighted in red both here and in the revised manuscript.

**Reply to referee #2**

In this manuscript, Li et al. conduct a comprehensive examination of how agitation methods, filter pore size, and contact time influence the solubility of nine aerosol trace elements through ultrapure water batch leaching techniques utilized across various labs. The findings indicate that the impact of these variations on solubility measurements is minimal, underscoring the necessity for standardized leaching procedures and the endorsement of ultrapure water batch leaching as a baseline protocol. Given the array of leaching protocols previously employed to extract dissolved aerosol trace elements, discerning the disparities among them is crucial. Overall, the article presents its results clearly, boasts a logical structure, and is well-written. Therefore, I recommend its publication once the minor issues listed below are addressed.

**Reply:** We would like to recommend ref #2 for reviewing our manuscript and recommending it for publication. We have addressed his/her comments which greatly help us improve our manuscript, and revised the manuscript accordingly, as detailed below.

The authors should briefly elucidate the rationale behind the selection of the nine elements for study in the introduction or methodology section. It is important to convey to the readers the representational significance of each element, ensuring that the selection appears deliberate rather than arbitrary.

**Reply:** In the revised manuscript (page 7) we have added one sentence to explain why we chose these elements: "These elements were chosen because they are important nutrients, toxic elements, or source tracers."

As the core of the article revolves around the comparison of analytical methods, the details of the methods should be more thoroughly delineated. (a) The authors mention collecting four sampling blanks and three laboratory blanks. They should elaborate on how these were utilized to correct the results and to what extent they affected the outcomes. Also, whether the analysis of blanks differs with the method should be detailed. (b) The use of ICP-MS for analysis is noted. It would be beneficial for the authors to introduce the QA/QC of ICP-MS analysis, particularly by

 information on the Method Detection Limits (MDL). Details on the ICP-MS used at NIO, if available, should be also included.

**Reply:** The blank levels were very low. In the revised manuscript (page 6) we have added one sentence to provide information for our blanks: "The amounts of dissolved trace elements on blank filters were mostly below detection limits; in a few cases the blank levels exceeded detection limits, but were negligible when compared to these on filter samples."

Information related to QA/QC and MDL can be found in original papers which described technical details. As a result, we do not repeat such information, but refer readers to relevant papers which we cite.

In Table 1, some cells are empty, which looks unprofessional. It is recommended to write 'NA' or use a dash line with a footnote for clarification. Additionally, what does 'lab' in the header signify? Is it a lab code or the location where the experiment was conducted?

Reply: In the revised manuscript (page 7) we have added one sentence in the table caption to explain what "lab" in the table means: "In this table, "Lab" represents the lab whose protocol was adopted in this work to digest or leach subsamples." In addition, as suggested, in the revised manuscript (page 7) we have filled empty cells with "--" when applicable.

Lines 184-185: A single sentence as a paragraph is inappropriate; it should be combined with the preceding section.

**Reply:** This paragraph only has one sentence, because we make this intentionally to underscore this sentence; as a result, we prefer to keep it as it is.

Table 2 displays t-test results, yet the discussion suggests that even with significant differences for certain elements, the discrepancy is minor. Thus, the importance of t-test outcomes seems diminished, and it is suggested to place them in supplementary materials.

**Reply:** We agree that Table 2 can be moved to SI. However, as the main text itself only have 3 tables and 4 figures, we decide to keep Table 2 in the main text.

In Figure 4, panels b and c, the authors should clarify the presence of two sets of equations (one black, one blue). Is it because one set excludes outliers? If so, how were these outliers determined? This should be clearly explained. Based on Figure 4(b), the presence of outliers appears to have a minimal impact on the linearity of the data.

**Reply:** In response to this comment, we have added the following sentence in the figure caption in the revised manuscript (page 17) to provide necessary explanation: "Black lines

represent fitting when all the data points are included, and blue lines represent fitting when outliers (represented by red crosses) are excluded." In addition, similar changes are implemented for figures in SI when necessary.

Minor Issues: (a) When describing the correlation coefficient, should it be capitalized as 'R' or 'r'? The manuscript uses both; please check and standardize throughout the text. (b). In Tables 1 and 2, some content is bold and underlined. While the intent to emphasize is clear, uniform formatting or an explanation in the footnote would be advisable. (c). The use of red to highlight information in the figure axis titles is unnecessary, as clarity is already achieved without it.

**Reply:** (a) In the revised manuscript we have changed all the "R" to "r", as suggested by the referee.

(b) We agree with the referee, and in the revised manuscript (page 7) we have added one sentence in the table caption for necessary explanation: "Experimental parameters for subsamples 2c-2e, when different from those for subsamples 2a, are highlighted in bold and underline."

(c) For these figures, x-axis is very similar to y-axis. As a result, we prefer to highlight the difference in red, as in this way readers can easily understand these figures.

---

## Author Comment (AC3)

Comments by referees are in blue.

Our replies are in black.

Changes to the manuscript are highlighted in red both here and in the revised manuscript.

**Reply to referee #3**

Li et al. used several ultrapure water batch leaching protocols to examine the effects of agitation methods, contact time, and filter pore size on the solubility of trace elements. This is good work and is recommended to be published in AMT after concerning the following weaknesses.

**Reply:** We would like to recommend ref #3 for reviewing our manuscript and recommending it for publication. We have addressed his/her comments which greatly help us improve our manuscript, and revised the manuscript accordingly, as detailed below.

Major comments: 1. Why is it needed to "formulate a standard operating procedure for ultrapure water batch leaching" when agitation methods, contact time, and filter pore size led to small or even insignificant differences in the solubility of trace elements?

**Reply:** There are two major reasons, and in the revised manuscript (page 19) we have added a few sentences to explain it clearly: "We note that large difference in solubility determined using the four common leaching protocol we examined was also observed for Fe and Al (Table 1); moreover, the experimental parameters examined in this work do not cover the whole ranges of these used by various ultrapure water batch leaching protocols used in previous studies. As a result, before a standard operating procedure can be formulated for ultrapure water batch leaching, the community will need to reach consensus on…"

2. In lines 246-253, the authors mentioned that longer contact time (2 and 4h) would cause an increase in solubility by 1.3 times for Zn and ~3.1 for As. However, why does this study only consider the contact time of 0.5-2h?

**Reply:** This is because the contact time is 2 h for the protocol used by GIG, and 0.5-1 h for these used by NIO, OUC and ZJU. In the revise manuscript (page 13) we have added one sentence to provide necessary explanation: "We examined the effects of these two contact time, as the contact time was 2 h for the GIG protocol, and 0.5-1 h for ZJU, OUC and NIO protocols (Table 1)."

3. More comparative experiments should be added to avoid uncertainties in each group for testing the distribution of aerosol particles and the effects of agitation methods, contact time, and filter pore size.

**Reply:** For each protocols listed in Table 1, 26 subsamples were examined, and the number for samples is quite large. In the revised manuscript (page 7), we have added one sentence in the caption of Table 1 to make this clear: "For each protocol, 26 subsamples were examined."

Minor suggestion. 1. Add the representation of the circles within each figure or in the first figure.

**Reply:** Each symbol (most circles in our manuscript) in these figures represent a data point. The *x*- and *y*-axises are clearly labelled for all the figures, and symbols are widely used in our community to represent data points; therefore, we feel it not necessary to state in figure captions what these symbols represent.